# Voluntary Assisted Dying and Community Palliative Care: A Retrospective Study in Victoria, Australia

**DOI:** 10.3390/nursrep15020034

**Published:** 2025-01-24

**Authors:** Robert Molenaar, Susan Lee, Jodi Lynch, Kelly Rogerson

**Affiliations:** 1Palliative Care South East, Narre Warren 3805, Australia; susan.lee@monash.edu (S.L.); jodi.lynch@palliativecarese.org.au (J.L.); kelly.rogerson@palliativecarese.org.au (K.R.); 2School of Nursing and Midwifery, Peninsula Campus, Monash University, Frankston 3199, Australia

**Keywords:** voluntary assisted death, palliative care, primary care, assisted suicide, nursing

## Abstract

**Background/Objectives**: Voluntary Assisted Dying in Australia was first legislated in 2019, with significant concern expressed by palliative care services about the impact on services. We aimed to describe the impact of Voluntary Assisted Dying on community-based palliative care client care. **Methods**: This study was a retrospective cohort study that compared the characteristics and outcomes of clients who expressed interest in VAD, those who chose voluntary assisted death, and the broader client population of the service. **Results**: Only 4% of the total client population expressed interest in VAD, and 1% died through VAD. Of the clients who expressed interest in VAD, most had malignancy as their primary diagnosis. The median length of palliative care service for clients who expressed interest in VAD was 101 days, compared to 48 days for all service clients. For 97% of individuals who died from taking the substance, death occurred in their place of choice compared with 71% of all service clients. Of the clients who died through VAD, 88% of these deaths occurred in a community setting compared with 56% of all service clients. **Conclusions**: Most clients who took the VAD medicine died in their place of choice, which was the community. A review of the length of engagement with the service indicated that a longer length of engagement was illustrated by individuals navigating the VAD process. This study emphasised the value of early referral to community-based palliative care, enabling a focus on quality of life, symptom management, and planning for death.

## 1. Introduction

The Voluntary Assisted Dying Act 2017 (Vic) was introduced in Australia as legislation in 2017 and first enabled in Victoria health services in 2019, amid significant concern and misunderstandings in the community about the perceived ‘inadequacy’ of palliative care [1] as presented in the media. The legislation provides for a person in the later stages of life-limiting illness to make a voluntary and enduring request for a prescribed substance to be provided by a doctor that will bring about their death, taken when they choose [2]. There are stipulations in the legislation regarding the training and permitting of providers, measures relating to the request and the control of the substance, and the prevention of health professionals raising the subject with patients with associated penalties. Health services, including palliative care services, were required to develop policies and procedures related to Voluntary Assisted Dying (VAD) in line with their service values, including appropriate training for staff.

The preparation of a community-based palliative care service to care for people seeking assisted dying has been reported as requiring an individual clinician response and an organisational response [3]. The literature to support this development has focussed on concerns about staff willingness to be involved, the challenges in the VAD process [4,5], and concerns regarding the protection of the wellbeing and legal rights of staff [6]. As the legislation was silent on the roles of nurses in the VAD process, there was no guidance for community services where clinical nurses provided assessment and ongoing care. Although it was anticipated that the number of referrals for VAD would be small [3], there were no studies to inform the education of staff that focussed on the proportion of clients and their characteristics, the length of time the process would take, and the possible outcomes of the process. Concerns were also raised as to what the potential financial, human, and resource costs associated with VAD would be [7] and there were calls for future studies to quantify the costs involved.

To inform the education and support of nurses and other health professionals and to assist community palliative care service delivery, we sought to describe the impact of VAD on community palliative care service delivery by examining a cohort of community palliative care clients who expressed their interest in VAD and their number, characteristics, and outcomes in comparison to those who had not, within the same service.

## 2. Materials and Methods

The setting of this study was a community palliative care service located in a metropolitan region of Victoria with a population size of 650,000 across 1800 square kilometres. Clinical staff from the service underwent external and internal education regarding what the VAD process involved before the VAD pathway was enabled. Only medical staff who choose to be involved in supporting VAD pursued specific VAD training. The population service data are recorded on an extensive data base and the use of this deidentified data for research is consented to on admission. The protocol was reviewed and approved by the Monash Health Research Support Service as a quality assurance activity (RES-23-0000-413Q), complying with the Australian National Health and Medical Research Council guidelines on these activities [8].

From 19 June 2019, when the legislation permitting VAD was enacted in Victoria, clients who indicated their interest were flagged and their progress through the various steps was documented in the service data base as a part of their ongoing care. A retrospective audit was undertaken to identify the demographic characteristics of clients who had died, where VAD had been mentioned in the record, and their progress through the VAD process from June 2019 to June 2023. These data were downloaded to a spreadsheet, and missing data were completed by reviewing the whole client record by author RM. Summary reports were generated on the whole client data base for the relevant demographic data fields (number of clients, sex, age, life-limiting condition, and length of stay) in addition to preferred site of death and whether that was met for the purposes of comparison. The data were checked for completeness and accuracy by author J.L. and then analysed by both authors independently using descriptive statistics.

## 3. Results

### 3.1. Characteristics of Applicants for VAD

During the study period, there were 121 clients of the service who sought information about or applied for VAD via the legislated medical process (VAD group), a total of 4% of the 3056 clients who had died in the service during the same period. Of these, 34 clients took the VAD medication, (VAD taken group), which equates to 1% of all service clients (Figure 1).

The demographic and baseline data of these three groups (Table 1) indicate that of the service VAD group, 59% were male, compared to 53% of the VAD taken clients and 51% of all those who died in the service. In the VAD group, the median age was 70 years, compared to 72 years in the VAD taken group and 77 years for all service clients.

In the VAD group, the primary life-limiting condition was a malignancy for 112 (93%) clients and a non-malignant diagnosis for 9 (7%) clients. In comparison, in the VAD taken group, 100% of clients had a malignancy; among all service deaths, 2136 (70%) had a malignancy and 920 (30%) had a non-malignancy diagnosis.

The median length of a service engagement with the service of the VAD group was 101 days compared with 117 days in the VAD taken group and 48 days for all service clients.

### 3.2. Location and Preferred Site of Death

Service clients died in one of four different locations (Table 2). The place of death for clients who had sought information about VAD was compared to the VAD taken group and all service deaths.

In the VAD group, 55% died in a private residence, 31% died in inpatient palliative care, 8% died in inpatient settings other than palliative care, and 6% died in residential aged care facilities. However, in the VAD taken group, 82% died in their private residence, 6% died in inpatient palliative care, 6% died in inpatient settings other than palliative care, and 6% died in residential aged care. Regarding all service deaths, 32% died in a private residence, 32% died in inpatient palliative care, 12% died in inpatient settings other than palliative care, and 24% died in residential aged care facilities.

The preferred site of death for all service clients was either met, not met, or unknown at the time of their death. In the VAD group, 70% had their preference met, 23% did not have their preference, and, for 7%, their preference was unknown, which was similar to all service deaths. In contrast to this, 97% of the VAD taken group’s preferred site of death was met and only 3% was not met. In the VAD taken group, 88% died in a community setting compared with 61% of the VAD group and 56% of all service deaths.

### 3.3. Characteristics of Service Deaths Attributed to VAD

Over the four-year study period, service deaths attributed to VAD substance administration have remained similar, with 7–9 deaths per year. Of the clients who had not used VAD, 10% of clients had the means of having the VAD substance administered but did not use it, even though available in their homes.

The VAD substance was either self-administered or practitioner-administered. In this service, 79% of VAD deaths involved self-administration and 21% were practitioner-administered.

## 4. Discussion

Clients of the service who died had a higher median age than the clients who showed interest in VAD. This is related to a larger amount of later-stage referrals that the service received from the community aged care sector. Many of these later-stage referrals are from clients who died within a residential aged care facility. This cohort of clients was more likely to have experienced cognitive decline and physical deterioration, which may have prohibited their ability to describe an interest in VAD or meet the requirements of the VAD assessment process [2].

A significantly higher portion of the service clients who died because of VAD died in their private residence or residential aged care facility, their preferred place of death, compared with those clients who expressed interest but did not die with VAD and with all other clients of the service. This difference could illustrate that those who died with VAD may have had more opportunity to plan and communicate the venue of care for the location of their death, whereas, for others, their illness trajectory towards the end of life often led to decisions being made to seek more supportive care away from their principal residence to either a palliative care unit or an acute hospital setting. This finding emphasises the value of explicit conversations about impending death, and the planning that includes preferences about the location of death.

The VAD taken group’s longer length of service engagement could be illustrative of this group’s capacity to plan. This group may include a cohort of clients with more stable disease who have the opportunity and time to reflect on the limitations imposed on their lifespan by disease and plan accordingly before they develop significant or sudden physical and/or cognitive decline. The length of service engagement emphasises the important relationship between VAD and palliative care, which has been a contested view [9]. In a recent publication, it was suggested that although some in palliative care view VAD as an important part of service delivery, others view the goals of palliative care as contrary to VAD [9].

However, both the VAD taken and VAD groups’ longer engagement with community palliative care services in this study provided an opportunity for more palliative care assessment, intervention, support, and care planning during their remaining lifespan. Longer engagement may enable these clients to make decisions that benefit them and their carers through the connection with supportive services.

Interestingly, a small group of clients who had completed the VAD pathway and had a means of having the substance administered did not have the VAD substance at the time of their death. It is possible that their deterioration was sudden, or that they exercised their choice not to have it. Likewise, in Victoria, of the VAD applicants who were issued with a permit for VAD substance, a minority did not have the VAD substance. The rate of self-administered VAD substance for the VAD group is similar to the Voluntary Assisted Dying Review Board applicants [10] but appears much greater than in other states, particularly Western Australia, which may be due to differences in legislation [11].

In Western Australia, since the initiation of VAD in 2021, the majority of participants had a practitioner-assisted administration of the VAD substance, either via the oral, nasogastric, or percutaneous endoscopic gastrostomy route or intravenous administration [12]. If a person is unable to independently prepare and ingest the substance or is concerned about their ability to undertake these actions, a practitioner is able to assist and administer the VAD substance, via any route, with the majority being intravenous [12]. However, Victorian legislation specifies oral administration by the client, which most clients utilise, and carers may assist in the preparation of the VAD substance. Approved Victorian medical practitioners are only permitted to assist and administer intravenous VAD substances if the person is unable to prepare or take the oral VAD substance [2].

Other differences in VAD legislation in Western Australia that may contribute to higher practitioner participation in VAD administration may be related to the ability of doctors and nurse practitioners to initiate VAD conversations with a patient, whilst, in Victoria, only patients can initiate VAD conversations. Also, nurse practitioners in Western Australia are permitted to administer the VAD substance, unlike nurse practitioners in Victoria [2,12].

Only 1% of all service clients had a VAD death recorded over the four-year period that VAD legislation has been enacted in Victoria. This is generally less than in other countries, where 0.3% to 4.6% of all deaths are reported as euthanasia or physician-assisted suicide in jurisdictions where they are legal [13]. For this service and for Victoria, the relatively small percentage of clients requesting information and those having the VAD substance is not reflective of concerns expressed elsewhere regarding a large increase in VAD requests following the implementation of VAD [14].

Although slightly more service clients sought more information about VAD, it is still a relatively small percentage of clients (4%) and did not overly impact the care service provision and resources available to be provided to all service clients. Despite the concerns expressed about the workload implications of the VAD process for doctors in Victoria, expressed by some authors [15], the small numbers are unlikely to impact palliative care services, although they tend to have a longer length of stay. The clients seeking VAD information were given general information about the VAD process by clinical staff face to face and contact details to dedicated VAD navigator services in Victoria, who provided more specialised support regarding VAD. The clients were then largely independent and autonomous in their engagement and linkage to the VAD process. If the client wished to have a clinical staff member present at the time of their taking the VAD medication, this was arranged with a clinician who was willing to be present at this part of the VAD process.

Clinical staff working in the community are generally experienced health practitioners and are also encouraged to complete further tertiary education in the palliative care field. They are thus accustomed to providing information to clients to help facilitate their death plans and support them in achieving their preferred end-of-life care venue. If a clinical staff member is not confident in providing this level of support, an internal referral to a more experienced clinician is made to assist the client in achieving the outcomes they wish for, and further training is provided to the clinical staff member involved.

This study represents the longest service study of Voluntary Assisted Dying in Australia but is limited by the small number of clients whose deaths could be attributed to this process. Further research should examine the impact of palliative care on the quality of life of people who seek information about VAD, in addition to following up on the financial and resource costs of VAD to community palliative care services.

## 5. Conclusions

This retrospective audit has illustrated that most clients who have taken the VAD substance have been able to die in their place of choice and have had a longer engagement with the community palliative care service.

An emphasis on continued conversations about planning for death, regardless of interest in VAD, is useful in determining earlier referral to palliative care, linkage to supportive services, and assisting other care decisions like place of death.

The number of service clients who wanted information on VAD was small and had little impact on overall community palliative care service provision, whilst the number of service VAD deaths that occurred over the four-year period was even smaller. There has been a low level of engagement by the entire client service group with the VAD process over this period since its inception.

Continued analysis of the service’s engagement with clients who seek information about VAD may illustrate future opportunities to improve the support that these clients could be offered as community palliative care services strive to achieve improvements in enabling their clients to die in their place of choice.

## Figures and Tables

**Figure 1 nursrep-15-00034-f001:**
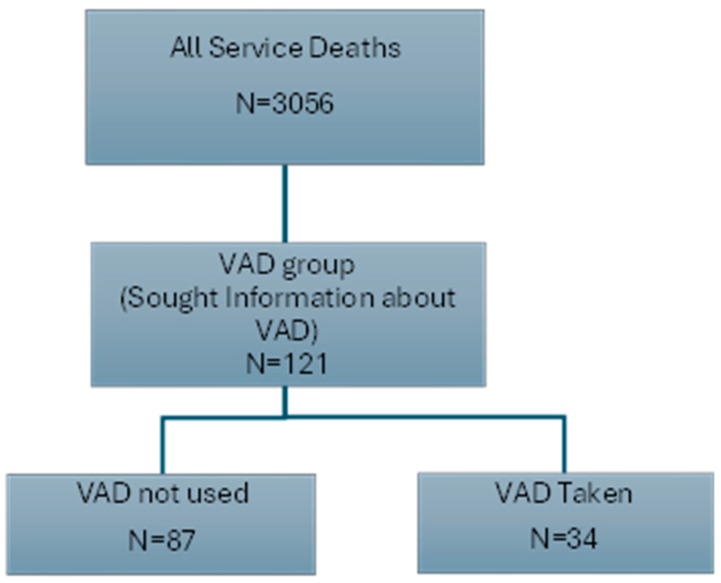
Deaths related to VAD in the service between 19 June 2019 to 30 June 2023.

**Table 1 nursrep-15-00034-t001:** Demographic and baseline data. Service information about VAD information seekers (VAD group), VAD taken group, and all service deaths.

Variables	Description	N (%) VAD Group	N (%) VAD Taken Group	N (%) All Service Deaths
Clients		121	34	3056
Sex	Female	49 (41%)	16 (47%)	1499 (49%)
	Male	72 (59%)	18 (53%)	1577 (51%)
Age (years)	Median	70	72	77
	IQR	61–76	59–78	67–86
Primary Life-Limiting Condition	Malignancy	112 (93%)	34 (100%)	2136 (70%)
	Non-malignancy	9 (7%)		920 (30%)
Length of Service Engagement (Days)	Median (IQR)	101 (40–223)	117 (47–241)	48 (13–133)

**Table 2 nursrep-15-00034-t002:** Location of death and preferred site of death in the VAD group, VAD taken group, and all service deaths.

Variables	Description	N (%)	N (%)	N (%)
Clients		121 (VAD group)	34 (VAD taken group)	3056 (All service deaths)
Location of death	Private Residence	66 (55%)	28 (82%)	980 (32%)
	Inpatient Palliative Care	38 (31%)	2 (6%)	966 (32%)
	Inpatient other than Palliative Care	10 (8%)	2 (6%)	377 (12%)
	Residential Aged Care	7 (6%)	2 (6%)	733 (24%)
Preferred site of death	Met	84 (70%)	33 (97%)	2169 (71%)
	Not met	28 (23%)	1 (3%)	656 (22%)
	Unknown	9 (7%)		231 (7%)

## Data Availability

The data presented in this study are available on request from the corresponding author due to the privacy and confidentiality of service data.

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
