# Peer review of "Voluntary Assisted Dying and Community Palliative Care: A Retrospective Study in Victoria, Australia"

_nursrep, 2025, doi:10.3390/nursrep15020034_

Round 1

Reviewer 1 Report

Comments and Suggestions for Authors

I found the article clear and comprehensible, and I think its value is also clear, in providing statistics useful for those working in the field, and also perhaps for the general public, to alleviate "tsunami" fears.  

The article notes  at the beginning that concerns were raised about the financial and human resource costs of the VAD program.  As this issue is raised, your reader is curious about those costs to the program studied.  Also, concerns regarding staff willingness to be involved in administering VAD are also mentioned, and perhaps these topics could be included in a follow-up study.  

I was curious about this statement: "The length of service engagement emphasises the important 146 relationship between VAD and palliative care, which has been a contested view."   Perhaps this is self-evident to the intended readership, but I would've valued one more sentence to explain the nature of the this contested view. 

I am a professor of Literature and a co-founder of the Program in Narrative Medicine at Columbia University.  My expertise does not lie in evaluating research design, but, again, I found the design clear and well conveyed. 

In that vein (narrative medicine) I will mention aspects of the article that I found worthy of further discussion, although, again, this is likely outside the objectives of the piece.

I was curious about the training and permitting of providers, how that training was delivered and how providers experienced it.   

The article urges more emphasis on conversations around death-planning, as the greater longevity of those who explored VAD, and their achieving their preferred location of death.   Is there training for those conversations?  should there be? 

One statistic that confused me a bit re: Preferred Site of Death:  This was MET by 70% of those who inquired about VAD, by 97% of those who took VAD, and 71% of all service deaths.  The figure for those who took VAD is  highest, I was a little surprised that the "all service deaths" figure was higher than those who had longer engagement with services.  

I have a small suggestions regarding wording: Opening sentence: 

 The Voluntary Assisted Dying Act 2017 (Vic) was introduced in Australia as legislation in 2017 and first enabled in Victoria health services in 2019, amid significant concern and misunderstandings in the community about the perceived ‘inadequacy’ of palliative 30 care [1] in the media 

I would add "due to representation in the media" or "as per..."  Something to that effect. 

And one typo in the final sentence: "with client’s who seek..."  Remove apostrophe.   

Reviewer 2 Report

Comments and Suggestions for Authors
  • Line 82: The figure caption "Deaths related to VAD in the service between June 19th 2019 to June 31st 2023." Should this be corrected to "June 30th, 2023?"
  • Line 87: "Table 1. Demographic and baseline data." Consider adding a brief description of the table's contents in the text before presenting the table.
  • Line 125: "Clients of the service who died had a higher median age than the clients who showed interest in VAD." This could be expanded to discuss why this might be the case and its implications (optional)
  • TITLE: consider specifying the geographic location (e.g., "in Victoria, Australia") to provide context.
